# Robust and Realtime Large Deformation Ultrasound Registration Using End-to-End Differentiable Displacement Optimisation

**DOI:** 10.3390/s23062876

**Published:** 2023-03-07

**Authors:** Mattias P. Heinrich, Hanna Siebert, Laura Graf, Sven Mischkewitz, Lasse Hansen

**Affiliations:** 1Institute of Medical Informatics, Universität zu Lübeck, 23562 Lübeck, Germany; 2ThinkSono GmbH, 14482 Potsdam, Germany; 3EchoScout GmbH, 23562 Lübeck, Germany

**Keywords:** ultrasound, image registration, deep learning, discrete optimisation

## Abstract

Image registration for temporal ultrasound sequences can be very beneficial for image-guided diagnostics and interventions. Cooperative human–machine systems that enable seamless assistance for both inexperienced and expert users during ultrasound examinations rely on robust, realtime motion estimation. Yet rapid and irregular motion patterns, varying image contrast and domain shifts in imaging devices pose a severe challenge to conventional realtime registration approaches. While learning-based registration networks have the promise of abstracting relevant features and delivering very fast inference times, they come at the potential risk of limited generalisation and robustness for unseen data; in particular, when trained with limited supervision. In this work, we demonstrate that these issues can be overcome by using end-to-end differentiable displacement optimisation. Our method involves a trainable feature backbone, a correlation layer that evaluates a large range of displacement options simultaneously and a differentiable regularisation module that ensures smooth and plausible deformation. In extensive experiments on public and private ultrasound datasets with very sparse ground truth annotation the method showed better generalisation abilities and overall accuracy than a VoxelMorph network with the same feature backbone, while being two times faster at inference.

## 1. Introduction

### 1.1. Motivation

Reliable realtime registration of ultrasound sequences can enable numerous practical applications in medical imaging. Tumours or organs-at-risk can be monitored and tracked in realtime to avoid unnecessary harm during radiotherapy or heat-based ablations [1], needle placement for biopsies or drug delivery can be assisted through image-guided navigation [2]. Image registration is also vital for freehand 3D reconstruction from manual ultrasound sweeps [3]. Diagnostic tasks, e.g., compression ultrasound for deep vein thrombosis (DVT) detection, that rely on dynamic human–machine interactions may be performed by frontline medical personnel with the help of realtime learning-based image analysis, instead of transferring patients to expert centres [4].

The most prominent previous study for ultrasound motion estimation in radiotherapy applications, the CLUST 2007 MICCAI challenge [1], focussed on tracking very few (four or less) pre-defined target anatomies over a period of time with mainly regular motion patterns. When adapting such algorithms to new ultrasound registration tasks, e.g., the evaluation of vein compression during realtime guidance for inexperienced users for DVT diagnosis, the rapid deformations and unpredictable motion can deteriorate the quality of trackers that rely on periodic motion. General purpose deep learning registration networks, such as PWC-Net [5] and VoxelMorph [6] may alleviate these problems and extend the applicability of ultrasound registration to new applications. However, they are prone to overfitting when using sparse supervision.

### 1.2. Related Work

The comprehensive medical multi-task medical registration challenge Learn2Reg [7] has shown that nearly all state-of-the-art deep learning-based registration tools, e.g., LapIRN [8], require strong supervision through densely annotated segmentation masks or keypoints. The classical optimisation-based approach of MEVIS [9] excelled at CT-based breathing motion estimation for which no manual expert supervision was available, but at the cost of very long run times of over a minute. PDD-Net [10] with metric supervision achieved the best learning-based results on multimodal ultrasound registration by employing mean-field inference, an idea borrowed from discrete optimisation, to regularise displacements. In related work from computer vision, Ref. [11] explored separable convolutions on a 4D cost tensor that reduced the degrees of freedom compared to the PWC-Net [5], but still required a large number of trainable parameters in this network part.

More specialised template tracking approaches, which rely on the presence of an annotated image in the first frame (template), include COSD-CNN (Cascaded one-shot deformable CNN) [12]. During training, template and instance images are processed by an unsupervised strategy to train a cross-correlation model that can roughly track the template. During inference a narrower crop is selected, based on the manual annotation and a one-shot deformable convolution module is used to fit the appearance transformation of the target structures in a self-adaptive fashion. Similarly, Ref. [13] incorporated on-line learning of a supporter model that captured the coupling of motion between image features, making it potentially useful for predicting target positions, which can be individually tracked. Further works, including [14,15,16], aimed to more explicitly incorporate temporal motion information through Conv-LSTMs, PCA motion models and a GAN-based Markov-like net that incorporates transformer modules respectively.

While basing the ultrasound registration on online learning for template appearance and/or a particular periodic motion model, helps to achieve high benchmark accuracies for datasets that fulfil these requirements, these tracking approaches may fail when unexpected motion, drift or tissue compression is present when considering a wider range of clinical applications. Furthermore, probe movement is not directly compensated for, leading to additional sources of errors, and external optical tracking devices are cumbersome in clinical practice.

Hence, there is still a need for a general purpose dense registration tool akin to the approaches that excelled at Learn2Reg, but with the exception of being trained with very sparse annotations. To alleviate the limited availability of expert annotations, a knowledge distillation framework for ultrasound registration was explored in [17], where a light-weight PDD-net was supervised by a PWC-net model trained on millions of natural image sequences. Unfortunately, the domain gap between popular computer vision datasets to clinical ultrasound appears to be too large for such approaches to yield high accuracy. An interesting alternative would be the consideration of unsupervised learning strategies from optical flow estimation in computer vision [18,19] that would, for example, employ synthetic occlusions to enable self-supervision on very large unlabelled datasets.

### 1.3. Contributions

In this work, we propose a combination of a straightforward feature backbone, together with a novel differentiable convex optimisation layer, that enables robust and accurate displacement predictions. This paper extends our original conference paper presented in [20] and an accepted abstract paper [21] where a preliminary version of our differentiable optimisation module was introduced, but only applied in a limited fashion to weigh and mix input channels from pre-trained or hand-crafted feature extractors [20] or only used with a limited evaluation on dense segmentations, respectively [21]. Here, an end-to-end learning of all convolutional layers is evaluated within a substantially bigger experimental validation, and with a particular focus on learning with sparse supervision, i.e., few manually tracked points in ultrasound sequences.

Our method works very robustly without online training without a-priori knowledge of templates or cropping regions and without any temporal analysis (i.e., on a frame-by-frame basis). A direct comparison with tracking algorithms was, therefore, not purposeful, so instead we evaluated the concept in comparison to other learning-based general purpose approaches, including VoxelMorph and fast optimisation with handcrafted features. Here, a number of data augmentation and unsupervised loss strategies were explored to help cope with limited variability and sparse labels in the training data.

It is very fast with more than 370 fps (or less than 3 ms per frame on a server GPU), which subsequently enables realtime use on mobile edge devices. The comprehensive experimental validation included two datasets: 10 sequences of ETHZ dataset from the public CLUST challenge [1] with 4284 image frames in total that comprises breathing motion of the liver and a private vein compression dataset with several hundred sequences (each with 21 frames) that are indicative for non-invasive DVT diagnosis in the groin [4].

## 2. Materials and Methods

In the following we describe, firstly, all methods that are devised for the task of learning-based dense ultrasound motion estimation. Secondly, we introduce the datasets, ablation studies, experimental and implementation details.

The main idea of our method was borrowed from our prior work, ConvexAdam, that ranked first in the comprehensive Learn2Reg MICCAI challenge series (2020–2021) [20]. In order to perform large-deformation image registration, it employs a fast GPU-based implementation of the coupled convex optimisation [22] that approximates a globally optimal solution of a discretised cost function for a densely sampled transformation. In contrast to our previous work, that was not aimed at realtime performance, we omit the time-consuming Adam-based instance optimisation. ConvexAdam has one important limitation, in that it does not enable backpropagation through the coupled optimisation, which is based on a non-differentiable argmin selection. This prevents end-to-end training and, hence, requires pre-trained or hand-crafted features that cannot be adapted to the task at hand. In the subsequent sections, we describe how a shared feature extractor for both input images can be effectively learned using a differentiable approximation of the coupled convex optimisation. As mentioned before, the dense correlation enables a more robust estimation, but in computer vision the PWC-Net [5] uses this complex within a highly complex architecture with millions of trainable parameters. Training the PWC-Net hence requires a much larger training dataset with a dense ground truth label, which is commonly not available in medical imaging.

Here we present, for the first time, a method that incorporates a fully-trainable ResNet feature backbone within an end-to-end differentiable discrete optimisation strategy and is supervised with only sparse supervision and without segmentation masks.

### 2.1. Methods

The conceptual overview of our approach is shown in Figure 1. Fixed and moving input grayscale images are fed into a siamese feature extractor (that comprises two identical ResNet18 networks with shared weights). The stride of the feature maps yields a four-fold spatial reduction, while the number of channels is increased to 64 dimensions. Next, a large correlation window of 9 × 9 voxels (equating to 33 × 33 pixels in original resolution) is computed using a correlation function that computes point-wise squared sum of differences of feature vectors across fixed and moving images. Finally, the proposed differentiable convex optimisation is employed to yield smooth displacements. It can be robustly trained in an end-to-end fashion using sparse supervision.

#### 2.1.1. Feature Backbone

We employed the ResNet-18 model as the feature backbone with the following moderate modifications. We removed the fourth block and prevented too aggressive downsampling by replacing the two-fold strides in the third block with normal convolutions (see Figure 2. The number of trainable parameters was reduced from 11 to 1 million, while the number of computations was slightly higher, with 5.74 vs. 2.22 billion Flops per image. We chose this model to balance network depth, to gain expressive feature abstraction to deal with challenging imaging artefacts, and to maintain high enough image resolution for the subsequent sub-pixel motion estimation.

#### 2.1.2. Differentiable Convex Optimisation

Our proposed approach is straightforward and well designed for realtime estimation of large motion for ultrasound sequences with temporal dynamics and is easier to implement because it does not require multiple warping steps, attention mechanisms or cascaded architectures.

The differentiable convex optimisation proposed here is an extension of a non-differentiable convex–discrete method [22], which aims to find a deformation field u by solving a cost function that simultaneously optimises smoothness (weighted by α) and feature similarity
(1)$$E\left(v,u\right)=CorrVol\left(v\right)+\theta {\left(v-u\right)}^{2}+\alpha {\left|\nabla u\right|}^{2}$$

Here, the displacement correlation volume CorrVol was computed from the above-mentioned ResNet features from fixed and moving images. Note that the correlation assigns a cost to each pixel and each potential displacement (here, 81) in a search window that slides over the moving image. This means an intermediate deformation field v can be obtained through the softmin operation, followed by an integral regression over all potential displacements. See also the concept of integral heatmap regression [23] for further details and the motivation of this last step. To encourage smoothness of the resulting transformation, an additional term is added to CorrVol by iterating a few times over Equation (Equation 1). The hyperparameter θ is increased after each step to adapt the coupling of similarity and regularisation penalty, which converts the non-convex optimisation problem into two coupled convex ones. This enables fast convergence to a global optimum in the space of potential displacements (which are predefined for the given 9 × 9 window) and, therefore, effectively avoids local minima. The coupling term models parabolas rooted at the current displacement estimation, updated by choosing the minimal cost solution in each step and providing a robust regularisation. To solve the second part of Equation (Equation 1) a spatial Gaussian smoothing to the previous displacement field is applied, implicitly solving Green’s function and, hence, α is indirectly defined through the Gaussian kernel. Crucially, we also incorporated a spatial B-spline smoothing filter at the end to estimate a plausible displacement field [20]. This complete module has no trainable parameters and requires less run time and memory than the feature extractor.

#### 2.1.3. Sparse Supervision

As mentioned before, our proposed approach excels when limited and sparse ground truth annotations are available for training. We rely on only a few (1–5) 2D landmarks in each frame of our training database. To enable the use of landmark supervision as drop-in replacement for spatial transformer loss functions, which rely on applying the full estimated deformation to a (multi-channel) label image, we created three concentric rings with increasing label number around each annotated 2D position (see example image in Figure 3) yielding three-class heatmaps around the manual landmarks. Subsequently, a standard soft Dice loss could be employed and the use of affine augmentation during training became easier. To improve the coverage of the estimated transform, and avoid overfitting of the features towards specific target anatomies that were annotated in the training dataset, we added the unsupervised MIND loss [24] with the following extension. Instead of using only 8 immediate neighbours to compute local contrast-invariant self-similarities, we sampled 64 2D offsets for patch comparisons that provided a better context representation. Finally, a loss on the standard deviation of the Jacobian determinant of the estimated displacements was used, motivated by the fact that this metric was used in Learn2Reg [7] as a quantitative smoothness measure.

#### 2.1.4. Extended Resnet–Voxelmorph Baseline

The second method, which we compared as a baseline and state-of-the-art benchmark, is an extension of the simplistic U-Net proposed in the VoxelMorph framework [6]. Different to the original version we did not feed the images directly into the VoxelMorph module, but employed the same ResNet-18 backbone as feature extractor. This is, hence, more similar to the successful approaches that proposed siamese tracking networks for ultrasound applications [25]. The final convolutional layer regresses a two channel displacement field, which is upsampled to the original image dimension using a B-spline function for improved smoothness, efficiently implemented as consecutive average pooling layers without stride (following the theory of recursive cardinal splines, as in [26]).

### 2.2. Datasets

#### 2.2.1. ETHZ

The public CLUST dataset https://clust.ethz.ch/data.html (acccessed on 6 May 2022) comprises several 2D ultrasound sequences of healthy volunteers freely breathing with multiple different scanners and transducers [1]. Here, we focused on ten sequences of several hundred frames each from the ETHZ dataset (see Figure 4). The temporal resolution was 15 Hz but higher frequencies were easily achievable with newer ultrasound probes. The image dimensions were approximately 400 × 400 pixels, with a square resolution of 0.4 mm. Either one or two anatomical landmarks were manually annotated by experts along each sequence to serve as both training objective and for cross-validation of the test accuracy. About a tenth of frames were labeled by three annotators to compute the inter-rater variance. The expected motion could reach several centimetres in each direction, i.e., motion of 16 pixels or more had to be estimated. So far, nearly all state-of-the-art approaches have used either temporal consistency (often coupled with online learning or fine-tuning) or restricted the search to a predefined template region. In our experiments, the whole image was used without cropping, and bilinearly downsampled to 160 × 192 pixels.

For many clinical applications, not only the location but the precise propagation of the contour or segmentation of an anatomical structure is required in each frame. In image-guided radiotherapy, organs-at-risk or a targeted tumour could deform during patient motion. In our second use case, the realtime guidance for inexperienced clinicians to perform accurate compression ultrasound analysis for deep vein thrombosis was considered. Here, the area of the investigated vein had to be monitored over the course of a manual compression to exclude a possible blockage that would point to an increased risk of a thrombosis clot and was potentially life threatening. During the examination, the vein had to kept within the field of view, despite very strong deformations, making the image interpretation particularly challenging. Providing a realtime guidance through overlaying an automatically propagated segmentation of the contour could be of great practical help and avoid user-dependent diagnostic errors [4]. Previous work relied on instance segmentation in each individual frame of the ultrasound video, which can lead to temporal instabilities, e.g., rapid switching between vein and artery label, sudden drop-outs (missing to segment a structure) and inaccuracies during compression. This would, hence, reduce user friendliness and highly prevent clinical adoption. Therefore, robust and fast image registration methods can help to improve the temporal continuity in the provided segmentation.

#### 2.2.2. CoCoAI

The second dataset, which was part of the *Cooperative and Communicating AI methods for medical image-guided diagnostics* (CoCoAI) project (https://www.imi.uni-luebeck.de/en/research/p46-cocoai.html, accessed on 21 February 2023), comprised 338 image sequences from handheld ultrasound showing one to five leg veins and arteries of healthy volunteers. All sequences showed a temporal view inside the leg at a diagnostically relevant anatomical landmark, where a vein was compressed by pressure applied on the probe, either to full disappearance or, in the case of a thrombus, to the size of the occlusion. We cropped all sequences to a common length of 21 frames and image dimensions of 160 × 160 pixels. While expert pixel-level segmentation were curated to measure registration accuracy. We used a simpler centre point annotation of each vessel to train our algorithms. This was much more scalable for the generation of larger datasets that, for example, cover further anatomical positions in future work. The compression also moved the background structures in a non-uniform pattern and caused changes in contrast and ambiguous vessel boundaries, making this a particularly hard registration problem.

## 3. Experiments

We performed a large number of experiments to evaluate the performance gains that result from replacing the popular VoxelMorph network for displacement prediction with our novel differentiable convex optimisation module.

Different to the participants of CLUST [1], we did not address the more restricted task of tracking a known template sequentially over time, but treated the alignment of the ETHZ dataset as a dense registration problem, where each pair within a sequence had to be registered without any temporal information—ETHZ dense unordered. This added challenge, but also yielded practical advantages. First, since we registered any pair of frames from a long temporal sequence regardless of any specific ordering, our approach could not learn to overfit on a particular breathing motion pattern and was, therefore, more robust against sudden changes and motion drift. Second, because our method predicts dense deformations, the template did not have to be known a priori and, due to the unsupervised MIND loss, all structures (including organs-at-risk) were registered accurately. For completeness we also evaluated our methods at inference time for a sequential ordering of frames **ETHZ dense sequential**, which was slightly easier, since the average misalignment was smaller and should result in higher accuracy, despite the fact that we still did not use temporal guidance. In each experiment, we performed a 5-fold cross validation with 8 training and 2 test sequences and ensured that the same patient was not in both sets at the same time (i.e., ETHZ-01-1 and ETHZ-01-2 had to be in the same set).

For the **CoCoAI** dataset for vein compression ultrasound [4], we performed the same setting, in which all images from one sequence were randomly drawn in pairs during training to avoid overfitting on simple incremental motion and deformation. The estimation of a dense displacement field was required to accurately not only detect translational, but also shape, transformations. As outlined before, we extracted a three-class heatmap around the manual landmarks to define our sparse loss. Different to ETHZ, we employed the Dice overlap based on 6’888 pixel-wise annotated frames with two classes (vein and artery) to evaluate the registration quality at test time. Note that this is more challenging since no supervision (in the form of pixelwise segmentation labels) is provided for training and the models need to, nevertheless, generalise well enough to align the object boundaries. The sequences showed substantial deformation due to compression with average initial Dice values of only 66%.

The selected two anatomies, liver and vessels of the legs, the range of employed ultrasound scanning probes, as well as the differences in motion, breathing and compression, already cover complementary challenges in realtime ultrasound analysis and, hence, the majority of our findings is very likely extendable to other body parts and imaging setups. Nevertheless, further research would, for example, be necessary for tracking motion and deformation in invasive ultrasound-guided biopsies etc.

### 3.1. Implementation Details

All methods and experiments were implemented using pytorch v1.10 (Cuda Version 11.5) and models were trained and evaluated with a single Nvidia RTX A4000. Each training was run for 3000 iterations with a batch-size of 8, that meant 4 epochs for CoCoAI and 8 epochs for ETHZ. We used the Adam optimiser with an initial learning rate of 0.001 and cosine and annealing scheduling with warm restarts every 375 iterations. When employing augmentation, a random contrast variation field using a B-spline with 20 × 24 control points and a geometric affine transformation with random standard deviation of 0.15 was used. MIND features were multiplied by 10 and used with an MSE loss, a soft-Dice loss for landmark heatmaps with $\epsilon =10^{-8}$ and a weighting of 2, as well as a Jacobian determinant standard deviation loss with weighting 0.5, were used as regularisation. The random augmentation and Jacobian loss were phased in with a sigmoid ramp-up to avoid early underfitting; in particular, for the VoxelMorph models. The regression output of VoxelMorph was obtained using a final hyperbolic tangent activation with multiplication of 0.2. For the differentiable convex optimisation we employed a grid-spacing of 4 and α=20 for the softmin operator. The coupling term θ was set to (0,0.3,1,3,10) and we found that employing only one iteration during training and five at inference gave the best performance.

To replicate our results for the public data and build upon the method we released open-source code at https://github.com/multimodallearning/differentiableConvex (accessed on 21 February 2023).

### 3.2. Ablation Studies

We performed numerous ablation studies on the **ETHZ** dataset to evaluate the influence of each of our contributions.

**MIND+convex:** First, we reimplemented a version of [15], which reached competitive results in the CLUST challenge, using handcrafted MIND features together with a PCA motion model and block-matching optimisation. Since we were here interested in a general purpose registration, we did not consider model-based or temporal regularisation but used only the proposed coupled-convex discrete optimisation with MIND features. This approach requires no training and, is, therefore directly employable for inference on unseen data.

**VoxelMorph:** Second, we used a standard VoxelMorph model [6] that used a direct concatenation of the two images as input and estimated the displacements using a U-Net. We used four levels and 64 channels as hyperparameters and an input size of half the image resolution. To improve smoothness of the transformations and ease learning with sparse supervision we appended a B-spline deformation model (see also Section 2.1.4). This constituted the fasted model.

**ResNet+VoxelMorph:** Next, we extended the VoxelMorph model by employing the ResNet-18 feature backbone, described in Section 2.1.1. This enabled a separate processing of both inputs in two streams and, hence, enabled a higher abstraction of features before concatenating them as input to the U-Net for displacement estimation.

**ResNet+convex:** Finally, we evaluated the proposed approach, which balanced complexity between the former two, because the differentiable correlation module was leaner than the U-Net and extended upon the first approach by introducing end-to-end trainable features. It also trained fastest among all learning-based methods.

**Augmentation:** In addition to the different choices in network design, we also evaluated the influence of input data augmentation. Due to the large inter-sequence variation in the ETHZ dataset and the small number of scans (despite a great number of frames per sequence) overfitting can become a serious problem. By adding both geometric affine transformations, that were applied to both input images and label maps the same way, and random field intensity augmentations the models could be made more generalisable.

**MIND loss:** While the sparse landmark annotations could already guide the models to register the most relevant features the remaining parts of the images could remain misaligned. Hence, the influence of the unsupervised MIND loss, that could be evaluated in the absence of spatially close landmarks, was also of interest.

**Capture range of correlation layer:** We explored whether the robustness of our approach could be further improved by using a larger window of potential displacements within the correlation layer and also set this to 13 × 13 (roughly twice as many displacements). We did not retrain the models that were optimised for 9 × 9. Please note that the required capture range depends on both the expected magnitude of deformation as well as the speed of the motion, which is studied in detail, as comparison between sequential and random ordering, in the next section.

## 4. Results

The quantitative results (see Table 1) showed large differences across the aforementioned methods. Considering the more challenging unordered pairwise evaluation of frames, we observed a reduction in target registration error (TRE) of 2.56 mm, or 48%, of our best approach (without MIND loss and capture range 13 × 13), compared to the baseline VoxelMorph. Even against the extended baseline with the same ResNet backbone our differentiable convex optimisation exhibited an improvement of 1.97 mm, or 42%. When considering the 90% percentile of errors, our approach reached 1.83 mm, which was 49% lower than the VoxelMorph baseline. The classic non-learning approach, that directly employs MIND features with convex optimisation, and not only indirectly through the loss function, came relatively close to our approach (+0.22 mm), based on the lower 90% percentile, and was 24% worse for the full set of landmarks. This indicated that sparsely supervised training could, indeed, improve the robustness of the motion estimation. A sequential evaluation, where a single central frame was defined as the template and all others were aligned towards it, resulted in expected improvement and the trend was similar, in that our approach was at least 47% superior to VoxelMorph.

Considering the ablation studies, the augmentation was found to be of clear importance, with more than 35% increase of error when omitting it, and removing the MIND loss led to a slight decrease in TRE for our approach, but a very substantial increase of 31% for the ResNet+VoxelMorph. This showed that VoxelMorph struggled particularly strongly with sparse supervision, while the intrinsic regularisation of our approach was more stable. The increased capture range of 13 × 13 led to a moderate improvement of 5%, most likely found for very challenging large deformations (and, hence, the lower 90th percentile error did not benefit).

Computational efficiency: The employed server GPU (RTX A4000) had a theoretical FP32 peak throughput of 38.35 TFlops when using mainly Tensor Core operations (counting multiply and add as two operations) and our best model reached 210 fps. This indicated a moderate utilisation of ≈17% of the peak performance. When translating this to a mobile device, e.g., a current iPhone with 3rd generation Neural Engine and 15.8 TFlops we expect 90 fps at FP16 precision, which would enable seamless integration into a user-friendly realtime guidance system.

Due to this efficiency, the training was also very fast, with 9 min per fold in our 5-fold cross validation. The UNet of VoxelMorph was trained twice as long, since it required longer to converge.

Finally, we also evaluated the two top performing models (from our proposed method and the VoxelMorph baselines) with one another on the **CoCoAI dataset**. As described above, the challenges of this dataset arise from strong deformation during vein compression, limited contrast of vessels and the potential of confusing artery and vein. Both models were trained with sparse supervision of only the centre points of each vessel across the 21 considered frames. For evaluation, we only used segmentation overlap by propagating the manual multi-class annotation between a fixed and moving frame using the estimated displacments. A comprehensive visual comparison of results for the ETHZ dataset of the two top models is shown in Figure 5. Table 2 shows the quantitative results for training on 300 sequences and testing on 38 held-out sequences. It became clear that the advantages seen in the previous experiments were also supported by a second dataset and our approach outperformed VoxelMorph by a large margin of 7.3% points, and reached a lower standard deviation.

We also investigated the influence of supervision level by using more detailed, dense segmentation labels, as well as no labelled data at all. Increasing supervision only led to a very minor improvement of 0.28% points Dice score and removing even sparse label supervision (and training solely with the unsupervised loss terms) only deteriorated the Dice score by 0.42% points. This finding further supports the fact that the intrinsic regularisation through differentiable optimisation of the proposed method reduces the reliance on expert labels. Conversely, the VoxelMorph baselines suffered a substantial deterioration in accuracy when trained without any labels, by 10% from 76.3%, and then became only marginally better than no registration at all.

In addition, we also trained an nnUNet [27] for individual frame segmentation. This required much more cumbersome annotation but could potentially help for difficult image quality at boundaries. Instead of aligning pairs of images the validation model was run in feed-forward mode and a direct Dice overlap could be obtained. Despite a slower computational speed (approximately 140 fps vs. ours with 210 fps) the segmentation approach was substantially less robust with 8% points lower Dice scores. That meant the registration-based guidance was likely more user-friendly and could improve the adoption in clinical practice.

The visual results are shown in Figure 6, along with a plot of the cumulative distribution of Dice accuracy across all 798 validation samples. The temporal averaging of all warped frames in Figure 6b) should ideally have a sharp mean. As expected, this was not fulfilled without registration. The VoxelMorph result also appeared blurry, while our approach found consistent mappings of images and labels. When considering the plausibility of the deformed segmentation, it appeared that, in contrast to our approach, VoxelMorph was not able to extrapolate from the sparse (centre) annotations to dense displacement that aligned all objects with their boundaries as well as the proposed differentiable convex optimisation.

## 5. Discussion

In this work, we presented a powerful methodological alternative for current state-of-the-art dense medical image registration techniques by combining our new differentiable convex optimisation module with an adapted ResNet-18 backbone for feature extraction. Contrary to our prior work [20], no dense segmentations are required to pre-train a feature extractor and, hence, our new approach can be learned in an end-to-end fashion from scratch on a wider range of problems. For inter-frame ultrasound registration, where the number of structures can vary across sequences, and often only sparse annotations of few centre points are available, the intrinsic regularising effect of the differentiable discrete optimisation incrementally adds a stronger penalty for non-smooth displacement fields to the correlation volume. Thus, more robust and accurate registration results are obtained. We do not assume any prior knowledge of specific targets or templates or periodic motion and, therefore, the algorithm can be seen as general purpose registration tool.

We employed two challenging datasets with thousands of ultrasound frames and performed extensive comparison experiments to baseline networks, classical methods and ablations of the proposed method. Our approach excelled in both applications and provided very low computational complexity with inference speeds of over 200 frames per second. A direct application on mobile devices without delay for user-friendly automatic guidance through our model can be obtained when assuming a current flagship phone or tablet.

In future, we would like to explore an extension of our method for a coarse-to-fine (multi-stage) motion estimation that could further improve precision. Pre-training with unsupervised data (through our image-based MIND loss) could further improve robustness across scanner types and motion patterns. Due to the flexibility of the models, we could also extend the evaluation to further modalities.

## Figures and Tables

**Figure 1 sensors-23-02876-f001:**
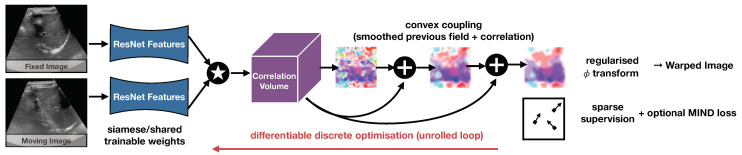
Concept of differentiable convex optimisation with ResNet feature backbone. The discrete correlation layer computes the dissimilarity across feature vectors of both images over a large search window in the moving scan. The optimum of the resulting correlation volume would yield a noisy deformation estimate that is smoothed. A coupling tensor that penalises deviations from the smooth estimate is added to the correlation in the following step yielding a refined smooth estimate. Through this intrinsic regularisation a very sparse supervision with 1–4 landmarks is sufficient.

**Figure 2 sensors-23-02876-f002:**
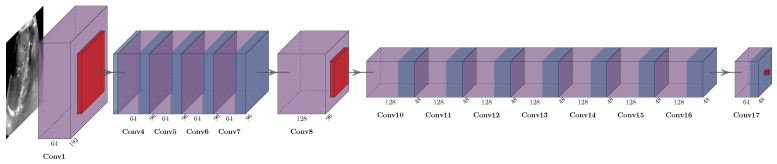
Visualisation of the modified ResNet-18 backbone for an example input image of 160 × 190, with reduced number of blocks, and, importantly, fewer downsampling steps to preserve spatial resolution of features. Residual connections were present in the model but not drawn.

**Figure 3 sensors-23-02876-f003:**
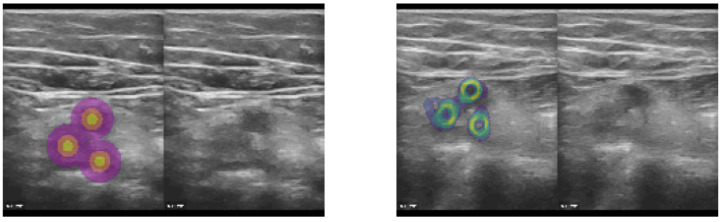
The concept was to employ sparse landmarks as spatial transformer (warping) loss by generating pseudo segmentation labels that could be directly used in affine augmentation and Dice loss functions. Based on either segmentation centroids or manual landmark coordinates (here vessel centres), we generated three concentric circles (heatmap labels) that defined the precise location, as well as a robust neighourhood of each landmark, for improved supervision. Left: Heatmap labels, along with original CoCoAI frame. Right: Tenth temporal frame in sequence with overlay of all warped heatmaps (middle label) averaged. It can be seen that the motion was very well compensated and only minimal deviations of the warped labels from their ground truth locations was visible.

**Figure 4 sensors-23-02876-f004:**
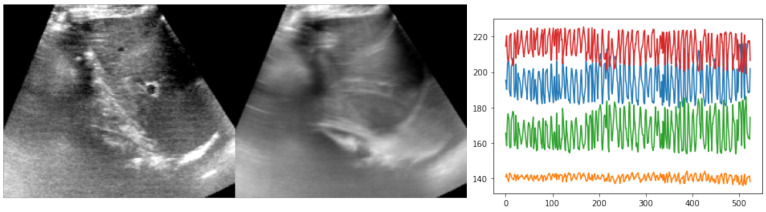
Example of reference frame and temporal average across one sequence from the ETHZ dataset, along with a plot of the observed motion, based on expert annotations (xy-coordinates for two landmarks). The temporal average, which showed substantial blurring of anatomies over time, as well as the large and rapid xy-coordinate changes in the annotations highlighted the difficulty of the registration task.

**Figure 5 sensors-23-02876-f005:**
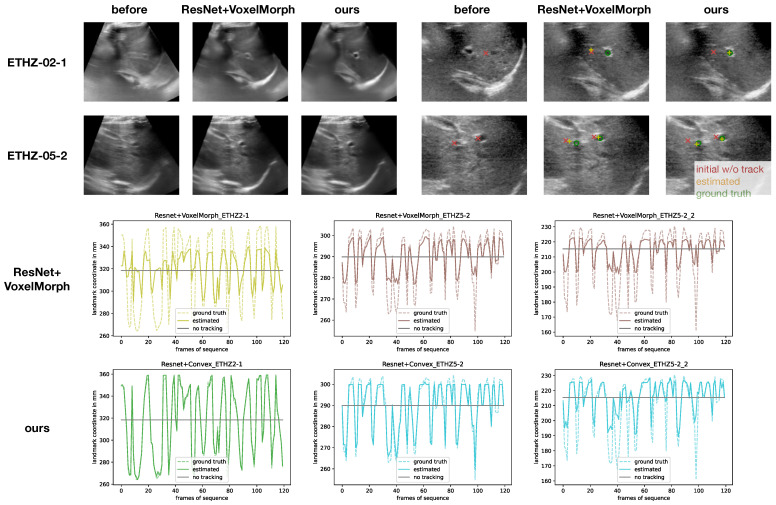
Comprehensive overview of selected qualitative results for the ETHZ dataset, comparing the best VoxelMorph method with our proposed method. The averaged warped images (over 120 frames) clearly demonstrate a very good motion compensation and sharp mean of our approach, whereas VoxelMorph’s result shows residual motion blur. The plots of tracking landmarks without temporal information show that VoxelMorph failed to capture larger displacement, while the convex optimisation excelled at sequences ETHZ-02-1 and ETHZ-05-2 for the first landmark and only slightly underestimated the sudden shifts in the second landmark of ETHZ-05-2.

**Figure 6 sensors-23-02876-f006:**
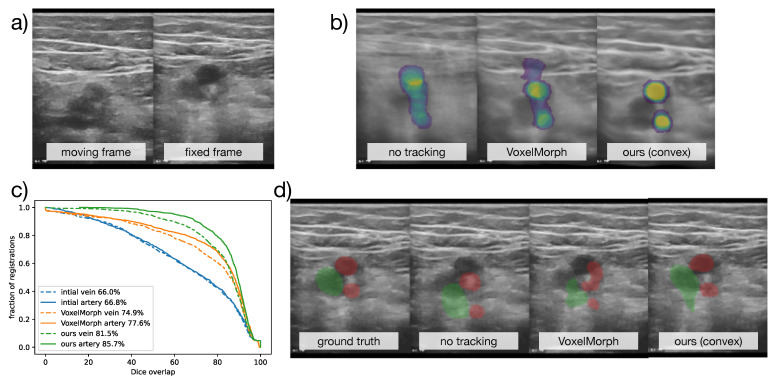
Qualitative results for the **CoCoAI** dataset. (**a**) shows two original frames from a compression sequence, (**b**) across all registrations with the same fixed frame the label probabilities for arteries after registration were averaged, and, ideally, two non-blurry circles should be seen, (**c**) the cumulative accuracy plot for all compared methods shows the advantage of our approach, (**d**) an example segmentation propagation (vein in green, arteries in red), shows that VoxelMorph’s deformation was not as plausible as ours.

**Table 1 sensors-23-02876-t001:** Quantitative evaluation of five-fold cross-validation of all approaches on the **ETHZ dataset**. Target registration error (TRE) is shown as an average in mm, as well as the 90th percentile (in brackets), based on initial misalignment. Best results per category are set in bold. Our approach was superior to all compared methods in terms of accuracy. It was much faster than the second and third best approaches and required the least amount of training time (except for the untrained MIND variant). Note that the first three columns all show the **unordered frame pairs** setup.

TRE in mm	w/o Augment	w/o MIND	Augment + MIND	Best Sequential	Train Time	Inference Speed
initial			6.84 (5.21)	5.04 (3.92)		
MIND+convex (13 × 13)			3.59 (2.05)	1.99 (1.40)	0 min	92 fps
VoxelMorph			5.28 (3.59)	3.34 (2.27)	12 min	**470 fps**
ResNet+VoxelMorph		6.14 (4.53)	4.69 (3.04)	2.80 (1.91)	18 min	280 fps
ResNet+convex (9 × 9)	3.35 (2.11)	2.86 **(1.62)**	3.09 (1.72)	1.54 **(1.21)**	9 min	305 fps
**ResNet+convex (13 × 13)**	3.72 (2.64)	**2.72** (1.83)	2.87 (1.87)	**1.48** (1.28)	9 min	210 fps

**Table 2 sensors-23-02876-t002:** Quantitative evaluation for the **CoCoAI dataset**. We compared the quality of segmentation propagation using motion estimation of our proposed registration approach against the best ResNet-VoxelMorph baseline. Both were trained with only centre point annotations of vessels. In addition, we also evaluated a segmentation model (nnUNet) that was trained on the full pixelwise annotation, but performed substantially worse, highlighting the advantages of registration for user-guidance.

Dice Overlap in %	Vein	Artery	Overall
initial	66.0 ± 12.5	66.8 ± 15.4	66.4 ± 13.4
nnUNet	72.06	78.96	75.51
ResNet+VoxelMorph	74.9 ± 11.6	77.6 ± 13.4	76.3 ± 12.0
**ResNet+convex (13 × 13)**	**81.5 ± 7.4**	**85.7 ± 7.1**	**83.6 ± 6.2**

## Data Availability

Algorithm raw performance results of this study are available from the corresponding author upon reasonable request. CoCoAI dataset: Image data access in line with the informed consent of the volunteer participants, subject to approval by the project ethics boards and under a formal Data Sharing and License Agreement on a case-by-case basis. CLUST dataset: access to data is possible after registration through https://clust.ethz.ch/data.html (accessed on 3 May 2022).

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
