# Peer review of "Robust and Realtime Large Deformation Ultrasound Registration Using End-to-End Differentiable Displacement Optimisation"

_sensors, 2023, doi:10.3390/s23062876_

Round 1

Reviewer 1 Report

The manuscript has proposed a new method and optimization algorithm for the realtime registration during ultrasonic image diagnosis, which is important in the healthcare field. Some questions need be addressed

(1) In the manuscript, the proposed ultrasound registration method were verified using two specific datasets including breathing motion of the liver and vein compression, which has a relatively fixed movement rhythm, so, in principle, is the method still effective when capturing images of other parts of the body with different dimensions and features during ultrasonic diagnosis? Does it need modification when used for diagnosing other parts of the body?

(2) The manuscript mentioned that the proposed method can be used for large deformation ultrasound registration, could you please explain in the manuscript how to avoid converging to local optimum? Whats the dimension range of the deformation that the method can be used for? In addition, does the movement speed influence the method for the registration process?

(3) The proposed realtime registration method involves deep learning, does it need large amount of data sample for training before it can be used for registration? If so, how much data is required?

(4) In 2.1.3 of the manuscript, the figure number is missing.

Author Response

A1) We thank the reviewer for the excellent comment and give further details about the comprehensive set of challenges that are already covered by our study as well as potential limitations e.g. for ultrasound-guided biopsies in the revised manuscript: "The selected two anatomies, liver and vessels of the legs, the range of employed ultrasound scanning probes, as well as the differences in motion, breathing and compression, already cover complementary challenges in realtime ultrasound analysis and hence the majority of our findings is very likely extendable to other body parts and imaging setups. Nevertheless, further research would e.g. be necessary for tracking motion and deformation in invasive ultrasound-guided biopsies etc."

A2) We thank the reviewer for the question and would like to point to "Differentiable convex optimisation" for details of who the coupled-convex optimisation enables the robust search for a regularised global optimum. We add the statement "and therefore effectively avoids local minima" to clarify this aspect further. In addition, we have amended the subsection "Capture range of correlation layer" with the following statement: "Please note that the required capture range depends on both the expected magnitude of deformation as well as the speed of the motion, which is studied in detail as comparison between sequential and random ordering in the next section."

A3) Indeed learning based methods do require a substantial amount of training data, 338 short sequences in the case of the CoCoAI dataset (see Section 2.2.2.), however our experiments also demonstrate that limited supervision, marking simply the centres of vessels instead of pixel-wise segmentation, is sufficient to obtain high quality results. Future work could investigate whether synthetic data augmentation or unsupervised learning can further reduce the data requirements. 

Reviewer 2 Report

Introducing a new form of ultrasonography image registration is evaluated.

For decreasing the uncertainties which are happened according to motion of living organs.

Author Response

We thank the reviewer for the positive remarks.

Reviewer 3 Report

In this manuscript, the author extended their previous study to propose a new method for deformation ultrasound registration. Comparison study was performed to demonstrate the performance. Results looks promising. I will recommend being accepted after addressing the comments below.

1.     Manuscript is not well organized. It should be further improved to make concise. Some figures are not well explained in the manuscript. In addition, Figure 5 wasn’t mentioned in the manuscript.

2.     Line 169: Please provide figure #.

3.     Provide scale bar for each ultrasound imaging.

4.     Provide detailed legends and captions will be helpful. For example, what different color represent in the right side of Figure 4.

Author Response

A1) We firstly would like to thank the reviewer for their positive general comments on our work. We have added more clarifications in numerous parts of the manuscript (highlighted in red) to address the concerns on conciseness.  We added the missing reference to the qualitative results figure: "A comprehensive visual comparison of results for the ETHZ dataset of the two top models is shown in Fig.~\ref{fig}.". 

We also extended the caption of Fig. 3, by adding: "Based on either segmentation centroids or manual landmark coordinates (here vessel centres), we generate three concentric circles (heatmap labels) that define the precise location as well as a robust neighourhood of each landmark for improved supervision." and "It can be seen that the motion is very well compensated and only minimal deviations of the warped labels from their ground truth locations is visible." 

Furthermore, we expanded upon the caption of Fig. 4 by adding: "The temporal average, which shows substantial blurring of anatomies over time, as well as the large and rapid xy-coordinate changes in the annotations highlight the difficulty of the registration task."

A2) We have added the missing figure number.

A3) The scaling of each ultrasound image depends on internal scanner parameter, e.g. gain settings, that are not available due to the anonymisation of the image data and hence the images are normalised by simple statistics (standard deviation) of observed grayvalues and the scales have to be considered arbitrary. 

A4) We thank the reviewer for the suggestion. Please refer to answer A1) in which we addressed the limited description of Fig. 3 and 4.  

Reviewer 4 Report

The paper presents a deep data-driven ultrasound registration method. A ResNet backbone is used to extract features from moving and reference frames. These are used in a correlation volume that represents the similarity given different pixel displacements within a pre-defined neighborhood/search window. This extends on previous work by formulating an iterative and differentiable optimization problem that balances the similarity between reference/registered image and a regularization term for promoting smoothness. Additionally, sparse supervision is provided through a few landmark annotations and a soft Dice loss. Results are evaluated on two datasets, in two formulations (unordered vs. sequential registration), and comparing to different configurations with manual and learned feature extractors and the VoxelMorph deep registration.

The method offers an efficient registration solution, and promising results are demonstrated. My main comments concerns the contribution over, and relation to, previous work. This goes both for the authors' previous work, as well as related work within unsupervised optical flow and cost volume formulation for correspondence matching between pixels.

Comments
---
* Contribution over previous work: The differentiable alignment optimization was presented in reference [17], but with pre-trained or manual features. A similar pipeline with learned features seems to be presented in reference [18], but with limited evaluation on dense segmentations. I would like to see a more thorough discussion in relation to the two previous work, to clarify exactly how this paper contributes over those. For example, from Section 2.1.2, it seems like the differentiable optimization is a contribution, but as far as I understand this has already been published.

* Relation to unsupervised optical flow: I am not that familiar with image registration in medical imaging. Thus, one question I have is how this is related to general optical flow methods, specifically unsupervised methods that can be used in a similar fashion as the correspondence matching, e.g.:
    -- Ren, Z., Yan, J., Ni, B., Liu, B., Yang, X., & Zha, H. (2017). Unsupervised deep learning for optical flow estimation. In Proceedings of the AAAI conference on artificial intelligence (Vol. 31, No. 1).
    -- Liu, P., Lyu, M., King, I., & Xu, J. (2019). Selflow: Self-supervised learning of optical flow. In Proceedings of the IEEE/CVF conference on computer vision and pattern recognition (pp. 4571-4580).

* Relation to previous cost volume methods: What are the similarities and differences compared to other cost volume based correspondence matching methods, e.g.:
    -- Yang, G., & Ramanan, D. (2019). Volumetric correspondence networks for optical flow. Advances in neural information processing systems, 32.

* Supervision evaluation: It would be interesting to see how the method performs with different levels of supervision (e.g. different number of landmarks used), and also completely unsupervised/self-supervised.

* Robustness evaluation: One motivation is that the proposed method should be more robust to variations in input data. It would be interesting to see some comparison in terms of generalization to data with a higher degree of domain shift, i.e. training on one dataset and testing on some other.

* Minor: undefined figure reference on page 5 (for fig. 3). Empty reference on page 7 (CoCoAI reference).

Author Response

We would firstly like to thank the reviewer for the very insightful and detailed comments on our proposed method and experiments. We are also thankful for the pointing out of further related works from computer vision.

Contribution over previous work: The reviewer correctly asserted that the work builds upon the two cited previous conference papers. We incorporate two clarifying citations to [17] and [18] at the respective parts of the contribution section that highlight the differences and modify the final sentence of the first paragraph as follows: "Here, an end-to-end learning of all convolutional layers is evaluated within a substantially bigger experimental validation and with a particular focus on learning with sparse supervision, i.e. few manually tracked points in ultrasound sequence.". Indeed the most significant contribution wrt. [18] is the extended experimental validation that investigates the influence of different loss terms and sparser supervision (see also comments below). In addition, we successfully experimented with a varying number of iterations in the coupled-convex optimisation between training and inference as described in "Implementation details", which is an improvement over [18]: "The coupling term θ was set to (0, 0.3, 1, 3, 10) and we found that employing only one iteration during training and five at inference gave the best performance."

Relation to unsupervised optical flow: We thank the reviewer for this very insightful suggestion. We have included the two citations and the following statement in the related work section: "An interesting alternative would be the consideration of unsupervised learning strategies from optical flow estimation in computer vision \cite{ren2017unsupervised,liu2019selflow} that e.g. employ synthetic occlusions to enable self-supervision on very large unlabelled datasets.". Currently, we believe the lack of sufficiently large unlabelled medical ultrasound datasets would prevent the direct adoption of deep optical flow models with several million parameters and hence believe the more restricted formulation of our learned feature extraction plus differentiable optimisation is beneficial in this case.

 Relation to previous cost volume methods: We thank the reviewer for pointing us to this very relevant related work. Yang & Ramanan’s method is similar to our own concurrently published PDD-Net in that it decouples spatial from "displacement" convolutions in the part of the network that follows the cost volume creation (based on fixed and moving features). It is, however, different to the submitted manuscript because this U-Net still imposes no intrinsic constraints for a regularised optimisation of those cost volumes and requires nevertheless a substantial amount of trainable parameters. We include the citation as follows:   "In related work from computer vision, \cite{yang2019volumetric} explore separable convolutions on a 4D cost tensor that reduces the degrees of freedom compared to the PWC-Net \cite{sun2018pwc}, but still requires a large number of trainable parameters in this network part."

Supervision evaluation: We thank the reviewer for this very constructive suggestion and have included additional experiments to address this open question. The following findings will be added to the revised manuscript: "We also investigate the influence of supervision level by using more detailed, dense segmentation labels as well as no labelled data at all. Increasing supervision only leads to a very minor improvement of 0.28% points Dice score and removing even sparse label supervision (and training solely with the unsupervised loss terms) only deteriorates the Dice score by 0.42% points. This finding further supports the fact that the intrinsic regularisation through differentiable optimisation of the proposed method reduces the reliance on expert labels. Conversely, the VoxelMorph baselines suffers a substantial deterioration in accuracy when trained without any labels by 10% from 76.3% and becomes then only marginally better than no registration at all." 

Robustness evaluation: We agree with the reviewer that cross-dataset evaluation would be of further interest. With only out-of-domain training on CLUST data our  model still outperforms the fully in-domain-supervised VoxelMorph baseline when both are evaluated on CoCoAI with 79.3% (a reduction of 4.3% against the in-domain setting) vs. 76.3%, which indicates a good robustness and improved generalisation. However, we believe that a more elaborate augmentation and domain generalisation strategy should be subject to further studies and we hope that this research direction will become more prominent within the whole medical imaging community. 

Minor: We have fixed both minor reference errors and thank the reviewer for pointing them out.

Round 2

Reviewer 3 Report

All the concerns have been addressed.

Reviewer 4 Report

In my opinion, the revision has sufficiently covered the aspects brought up in the previous review round. I appreciate the clarification in relation to the authors' previous work, and related work for self-supervised optical flow estimation and cost volume formulation. Also, it is interesting and promising to see that the method can be used in a completely self-supervised setup with only a small reduction in performance.